# *Rhizobium* Inoculation Improved the Rhizosphere P Dynamics and P Uptake Capacity of Pigeon Pea Plants Grown in Strongly Weathered Soil Only under P Fertilized Conditions

**Saki Yamamoto** [1], **Shin Okazaki** [1], **Nakei D. Monica** [2], **Naoko Ohkama-Ohtsu** [1], **Haruo Tanaka** [1] **and Soh Sugihara** [1,*]

[1] Institute of Agriculture, Tokyo University of Agriculture and Technology, Saiwaicho 3-5-8, Fuchu 183-8509, Tokyo, Japan

[2] Department of Soil and Geological Science, Sokoine University of Agriculture, Morogoro P.O. Box 30007, Tanzania

[*] Correspondence: sohs@cc.tuat.ac.jp; Tel.: +81-42-367-5676

**Abstract:** The improvement of phosphorus (P) use efficiency (PUE) is a critical problem in crop production because of phosphorus' scarcity. Especially in strongly weathered soil with a high P fixation capacity, a low PUE generally limits plant growth. Here, in a 70-day pigeon pea cultivation pot experiment using Ultisols, we evaluated the effects of *Rhizobium* inoculation ($-$I/+I) on the plant growth, rhizosphere, bulk soil P dynamics, and plant root P acquisition characteristics, with or without P fertilization (0P: no P application; 50SSP:50 kg P ha$^{-1}$ with single superphosphate). The combination of *Rhizobium* inoculation with P fertilization (50SSP + I) increased the plant growth, P uptake, and organic acid content per pot by 63%, 41%, and 130%, respectively, but not without P fertilization (0P + I). The labile and moderately labile inorganic P (NaHCO$_3$-Pi and NaOH-Pi) contents were higher (55% and 44%, respectively) in the rhizosphere soil than those in the bulk soil in the 50SSP + I treatment, indicating the efficient solubilization of the applied P under the 50SSP + I treatment. The fertilized PUE was higher in the 50SSP + I treatment (26%) than that in the 50SSP$-$I treatment (15%). Thus, these results suggest that *Rhizobium* inoculation with 50SSP should stimulate plant root P acquisition characteristics, leading to the solubilization of applied P in the rhizosphere and efficient plant P uptake. In conclusion, the 50SSP + I treatment effectively improved the PUE of pigeon peas in strongly weathered soil.

**Keywords:** *Rhizobium* inoculation; phosphorus use efficiency; rhizosphere; organic acid; Ultisols

## 1. Introduction

Phosphorus (P) is one of the essential elements for plants and supports many functions necessary to sustain life, including genetics, energy metabolism, and homeostasis [1]. However, its low mobility in soil results in poor uptake by plants, which negatively affects crop growth and yield [2]. Moreover, the resources of P are limited and rapidly decreasing worldwide [3,4]; thus, it is strongly necessary to improve the fertilized P use efficiency (PUE) to attain sustainable crop productivity [3]. In strongly weathered tropical soils with high P-fixing characteristics, the PUE is generally low due to the high contents of Fe and Al oxides in the soil [5,6]. These studies indicate that P limitation can be one of the major constraints of crop production in such areas. Therefore, improving the soil–plant P dynamics is essential to achieving sustainable productivity by improved P fertilization management in strongly weathered soils [7–10].

In recent decades, many researchers have discovered some unique abilities of P-efficient legumes; some legumes can solubilize and uptake moderately labile P as well as labile P in soils by secreting organic acids and phosphomonoesterase [11–14]. The

organic acids released from the plant roots change the pH of the rhizosphere soil, solubilize Fe/Al oxides and precipitated Ca, or activate the microorganisms in the soil, which consequently changes the P concentration in the rhizosphere soil and increases the plant's P availability [12]. Acid phosphomonoesterase secreted by the roots of several crop plants, including legumes, increases the plants' available P with hydrolyzing organic phosphorus (Po) [15,16]. Several legumes, such as pigeon pea (*Cajanus cajan L.*), groundnut (*Arachis hypogaea L.*), and white lupin (*Lupinus albus L.*) have been widely studied as P-efficient legumes [17–19]. In the tropics, pigeon pea is one of the most popular P-efficient legumes; it secretes many organic acids, which facilitate the absorption of moderately labile P even in heavily weathered soils [17,20].

To increase the crop growth or yield of P-efficient legumes in weathered soils, many studies have focused on P fertilizer applications [21,22], *Rhizobium* inoculation [23–25], and combinations of these methods. The combined application of *Rhizobium* inoculation and P fertilization has improved the crop yield and plant growth parameters in several crops compared to their separate applications [26–28]. For example, the combined application of *Rhizobium* inoculation and P fertilization increased chickpea yield ~1.7 times in Nigeria [29]. Ronner et al. [26] also reported a similar effect of combined application in soybeans in Nigeria. These results suggest that *Rhizobium* inoculation with P fertilization could improve the PUE of legumes; however, these studies did not evaluate the detailed soil P dynamics or plant P acquisition mechanisms.

Generally, legume–*Rhizobium* symbiosis requires a certain amount of P uptake for better $N_2$ fixation by the rhizobia [30,31]. Therefore, the expected mechanisms for better PUE by the combined application of *Rhizobium* and P fertilization are: (1) *Rhizobium* inoculation with P fertilization for legumes first improves the *Rhizobium* $N_2$ fixation capacity; and (2) it then improves plant growth by improving the P acquisition characteristics of plant roots, such as improved root mass or exudate. Rondina et al. [32] demonstrated that *Rhizobium* inoculation improved plant P uptake with increments of the root surface area, while Umali-Garcia et al. [33] found similar results caused by increased root biomass. However, there is limited information about root P acquisition characteristics by root exudate, i.e., root organic acids and phosphatase activity, as well as related rhizosphere soil P dynamics under *Rhizobium* inoculation with P fertilization. Therefore, a comprehensive study of plant root P acquisition characteristics and rhizosphere P dynamics in strongly weathered soil could reveal the above mechanisms and facilitate better PUE by their combined application. This could also explore the potential applications or limitations of the technique in the tropics.

Therefore, this study aimed to identify the effects of *Rhizobium* inoculation with P fertilization on: (1) the P acquisition characteristics of plant roots, such as root mass, root organic acid, and phosphatase activity; (2) the rhizosphere soil P dynamics; and (3) the plant P uptake and PUE in strongly weathered soil. The results of this study may serve as a valuable tool for improving PUE and achieving sustainable crop productivity in the tropics. Here, we hypothesized that *Rhizobium* inoculation with P fertilization improves plant root biomass and its exudate, i.e., plant P acquisition characteristics, resulting in greater P availability in the rhizosphere and, thus, improving the PUE. To verify this hypothesis, we conducted a pot experiment with two treatments, that is, *Rhizobium* inoculation and P fertilization, and analyzed the plant root exudates for both organic acid and phosphatase activity, rhizosphere soil P by the Hedley sequential extraction method [34], and plant P uptake.

## 2. Materials and Methods

### 2.1. Soils

The soils for cultivation were collected from the uncultivated surface layer (0–15 cm) of Ultisols (Soil Survey Staff, 2014) in the Okinawa prefecture of Japan (26°41′ N and 128°06′ E), which had not been fertilized and no crops had been planted there before. They were passed through a 4 mm sieve after being air-dried and used for cultivation. The soils'

physico-chemical properties are shown in Table 1 and include pH (1:5 water) 4.2, available P estimated via the Truog method [35] 10.4 mg P kg$^{-1}$, total organic carbon 25.1 g C kg$^{-1}$, total nitrogen 1.9 g N kg$^{-1}$, and clay 24%.

**Table 1.** Soils' physico-chemical properties.

| | Soil pH (H$_2$O) (KCl) | | Sand | Silt (%) | Clay | TC (g kg$^{-1}$) | TN | Total-P (mg kg$^{-1}$) | Truog-P | P Ads. Cap. (g P$_2$O$_5$ kg$^{-1}$) | Al$_o$ | Fe$_o$ (g kg$^{-1}$) | Al$_d$ | Fe$_d$ |
|---|---|---|---|---|---|---|---|---|---|---|---|---|---|---|
| Ultisols | 4.2 | 3.3 | 40 | 36 | 24 | 25.1 | 1.9 | 349.1 | 10.4 | 6.6 | 1.2 | 1.3 | 2.9 | 29.2 |

P Ads. Cap. is P adsorption capacity of soil. Al$_o$ and Fe$_o$ are oxalate-extractable Al and Fe, respectively. Al$_d$ and Fe$_d$ are dithionite-citrate-bicarbonate extractable Al and Fe, respectively.

### 2.2. Plant Growth Conditions

We conducted plant cultivation in an enclosed glasshouse (a phytotron) under natural sunlight with controlled humidity (60%) and temperature (25.0–18.0 °C, day 16 h—night 8 h regime). We placed 1.0 kg of soil into each pot (1/10,000 a) and kept soil moisture content at 60% of field capacity. We used pigeon pea, a legume crop widely cultivated as an intercrop species in tropical, highly weathered soils [36]. The pigeon pea seeds were surface-sterilized with sodium hypochlorite (1%) and ethanol (70%), wrapped in wet paper, and placed in a petri dish for 2 days at 25 °C for germination. After germination, the selected rhizobial solution, as explained below, was applied to their radicle to inoculate it with *Rhizobium* (1.7 × 10$^9$ cell per seed). One day later, four seeds per pot were planted, and rhizobia solution (5.0 mL) was applied again to the soil where the seeds were sown (3.0 × 10$^9$ cell per seed). The plants were thinned to two per pot 10 days after planting.

The inoculated rhizobial species for pigeon pea were selected by screening tests. Based on the previous report by Fossou et al. [37] which stated that *Bradyrhizobium* species are well isolated from pigeon peas, screening tests were conducted using the following five *Bradyrhizobium* species: *Bradyrhizobium elkanii* USDA61 (USDA), *Bradyrhizobium diazoefficiens* USDA110 (USDA), *Bradyrhizobium japonicum* USDA6 (USDA), *Bradyrhizobium japonicum* NKS4 [38], and *Bradyrhizobium* sp. DASA2007. After 30 days of screening experiment, USDA61 had the highest above-ground and below-ground biomass (Figure S1), and thus we selected it for the pot experiment. Before the inoculation, USDA61 was cultured in AG medium [39] on a shaker at 180 rpm and 28 °C for three days (3.0 × 10$^7$ cell mL$^{-1}$). After that, it was washed and diluted with distilled water to adjust the concentration as mentioned above.

### 2.3. Experimental Design

We conducted the experiment in a completely randomized block, including *Rhizobium* inoculation and P application treatments, with five replicates. *Rhizobium* inoculation treatment was divided into two categories: without *Rhizobium* inoculation (−I) and with *Rhizobium* inoculation (+I). For P fertilization, single superphosphate (SSP) was mixed with the soil sample two days before planting. The P fertilization treatments comprised no P application (0P) and 50 kg P ha$^{-1}$ P application (50SSP). In summary, four treatments were used in this study: 0P + I (only *Rhizobium*), 0P−I (no P, no *Rhizobium*), 50SSP + I (P + *Rhizobium*), and 50SSP − I (only P) (Figure S2). Furthermore, as the rhizosphere soil sampling may cause cellular damage to the root surface, causing overestimation in the root exudate analysis, two sets of pot experiments were conducted in this study to separately analyze plant root exudates (organic acids and acid phosphatase) and rhizosphere soil P dynamics. One set was used to assess plant biomass, N/P concentration, and rhizosphere soil P dynamics. The other set was used for root exudate analysis (Section 2.6). Thus, the total number of pots was 40 (2 *Rhizobium* inoculation × 2 P fertilization × 5 replicates × 2 sets = 40 pots). We mixed the soil with basal nutrients (g kg$^{-1}$ soil): MgSO$_4$·7H$_2$O 0.39 and K$_2$SO$_4$ 0.17, and with micronutrients (mg kg$^{-1}$ soil): ZnSO$_4$·7H$_2$O 2.2, CuSO$_4$·5H$_2$O 2.0, MnSO$_4$·5H$_2$O 0.6, H$_3$BO$_3$ 0.5, Na$_2$MoO$_4$·2H$_2$O 0.5, CoCl$_2$·6H$_2$O 0.4, and FeNa-EDTA 0.4 [40]. The pH was adjusted to 5.5 using Ca (OH)$_2$. Planting was performed in October

2020, and plants and soils were sampled after 70 days in December 2020. In this experiment, we considered that the difference between rhizosphere soil and bulk soil indicates the cumulative effect of P uptake by plants for 70-day cultivation period, while root exudate analysis indicates the potential of plant P uptake at harvest, which accounts for the difference in rhizosphere and bulk soil P dynamics.

### 2.4. Plant and Soil Sampling

The above-ground part (leaf and stem) was cut off at the soil's surface, and its root was washed well after collecting the rhizosphere and bulk soil (further details below). We also collected and measured the number and total weight of *Rhizobium* root nodules for each pot by hand-picking from the root system. All plant parts were oven-dried at 70 °C for more than 48 h and weighed for each part. The P concentration was evaluated calorimetrically using the molybdate/ascorbic acid method following nitric and sulfuric acid digestion [41]. The N concentration was measured using dry combustion methods with an NC analyzer (Sumigraph NC-TR22, Sumika Chemical Analysis Services Ltd., Osaka, Japan). The rhizosphere soils, which are root-adhering soils and affected by plant roots, including their exudates, were collected by shaking the root system in a plastic bag, and bulk soils were collected from other soils (not affected by plant roots).

### 2.5. Phosphorus Fractionation of Soil

The modified Hedley fractionation method [34,42,43] was conducted with the following four steps: (1) Resin-P, distilled water with two resin strips; (2) $NaHCO_3$-P, $NaHCO_3$ pH 8.5 (0.5 M); (3) NaOH-P, NaOH (0.1 M); and (4) HCl-P, HCl (1 M). Each step included horizontal shaking for 16 h at 25 °C and centrifugation ($2210\times g$) for 20 min. The content of inorganic P (Pi) was evaluated with the Murphy and Riley colorimetric method [44]. For the determination of total P (Pt) in $NaHCO_3$-P and NaOH-P, dissolved organic matter was digested with ammonium persulfate and $H_2SO_4$ (0.9 M) in the autoclave and measured as well as Pi; the calculated Po was the difference between Pt and Pi. Some of the fractions are classified as follows: labile P forms (Resin-P and $NaHCO_3$-P) and moderately labile P form (NaOH-P).

### 2.6. Evaluation of Exudation from Root

2.6.1. Evaluation of Organic Acid from Root

Root organic acids were extracted in distilled water, as described by Lipton et al. [45], Von Wirén et al. [46], and Sugiura et al. [47]. We collected all root samples with minimal disturbance and less damage to the entire root system, including fine roots, by immersing the entire cultivation pot in a large bucket of water and collecting the entire root system. After collecting, we carefully removed the whole pot of soil with plants and gently washed it in a large bucket with a large amount of tap water to completely remove the adhering soils with as little damage as possible. Then the plants were washed and rinsed three times with Milli-Q water. All parts of the washed root samples were soaked and kept in 200 mL Milli-Q water for 2 h at 25 °C to collect the exudation from roots. After that, the extracted solutions were stored at −20 °C and with lyophilize. The lyophilized samples were filtered with a sterile filter unit (pore size: 0.22 μm) to remove impurities following the dissolution in Milli-Q water. After that, organic acids in the samples were analyzed with a high-performance liquid chromatography system under the same conditions as those of Sugiura et al. [47]: a column temperature of 40 °C, analysis time of 50 min, a flow rate of 0.5 mL/min, and a mobile phase of $H_3PO_4$ [0.1% (*v/v*)]. The five kinds of standards of organic acid (oxalic, malonic, succinic, malic, and citric acid) were also analyzed after the adjustment to proper concentration.

2.6.2. Evaluation of Acid Phosphatase Activity of Root

After extracting the organic acid samples, we measured the fresh weight of the whole root to calculate the amount of exudation per root biomass and then used the same root

samples to evaluate the acid phosphatase activity of the root, according to the methods of Ishidzuka [48]. We used phosphate monoester p-nitrophenyl phosphate (pNPP) as a substrate, and buffer solution was adjusted to pH of 5.5 based on pre-experiment (data not shown). After incubation of root strip samples (including some control and correction samples), the absorbance of their supernatant was measured at 410 nm with a spectrophotometer.

### 2.7. Data Analysis

We conducted statistical analyses using SYSTAT 12.5 (SYSTAT Software, Richmond, CA, USA). We conducted one-way analysis of variance (ANOVA) to detect significant differences among the rhizosphere and bulk samples in each P fraction and plant-related values in each treatment. The means showing a significant difference in ANOVA were compared with posthoc Tukey multiple comparison tests. Three-way ANOVA was conducted to evaluate the effect of *Rhizobium* inoculation, P fertilization, and rhizosphere on soil P fraction. A two-way ANOVA was performed to evaluate the effect of *Rhizobium* inoculation and P fertilizer on the amount of organic acid and acid phosphatase activity. For all data, $p < 0.05$ was considered significant in all cases, and expressed on a dry weight basis.

To compare fertilized PUE, the P fertilizer recovery efficiency for each pot was calculated as follows [49,50]:

$$\text{\% fertilized P use efficiency (PUE)} = \text{\% P recovery of P fertilizer}$$
$$= ((U_p - U_0)/F_p) \times 100 \tag{1}$$

The abbreviations included in the above calculation and their definitions are as follows: $U_p$, the total plant P uptake under 50SSP treatments; $U_0$, the total plant P uptake under 0P treatments; and $F_p$, the amount of given P per pot.

## 3. Results

### 3.1. Plant Biomass, Plant P and N Uptake, and PUE

The plant biomass, P uptake, and N uptake of pigeon pea plants after the pot experiment are shown in Figure 1, and the P and N concentrations are shown in Table S1. A summary of the effect of the *Rhizobium* inoculation and P fertilizer on the plant biomass, P uptake, and N uptake is shown in Table 2. P fertilization increased the plant biomass, P uptake, and N uptake by 126%, 121%, and 38%, respectively. The plants' P concentrations in each part (leaves, stems, and roots) were not significantly different between 0P − I and 50SSP − I, while the plant N concentration under 0P−I was higher than that of 50SSP − I for the individual parts. The *Rhizobium* inoculation increased the plant biomass, P uptake, and N uptake under the 50SSP treatment by 67%, 41%, and 41%, respectively, whereas no significant differences were observed under the 0P treatment. The total weight of the root nodules (g pot$^{-1}$; the fresh weight) was 0.02 (0P − I and 0P + I) < 0.31 (50SSP − I) < 0.80 (50SSP + I), and the total number of root nodules also followed a similar trend (data not shown). These results indicate that P application clearly increased the root nodule number and weight, while *Rhizobium* inoculation in the 0P treatment did not. In addition, we visually confirmed that the nodule color of 50SSP − I was mostly grey or green, indicating inactive nodules for N$_2$-fixation, while the nodule color of 50SSP + I was mostly red, indicating active nodules for N$_2$-fixation. The P concentration in the leaves, stems, and roots of −I was higher than that of +I under both 0P and 50SSP, except for the roots under the 0P treatment and the leaves under the 50SSP treatment. The N concentration in the leaves, stems, and roots did not change, except for the stems under the 50SSP treatment. Finally, the PUE in 50SSP + I (25.5%) was higher than that in 50SSP − I (14.5%).

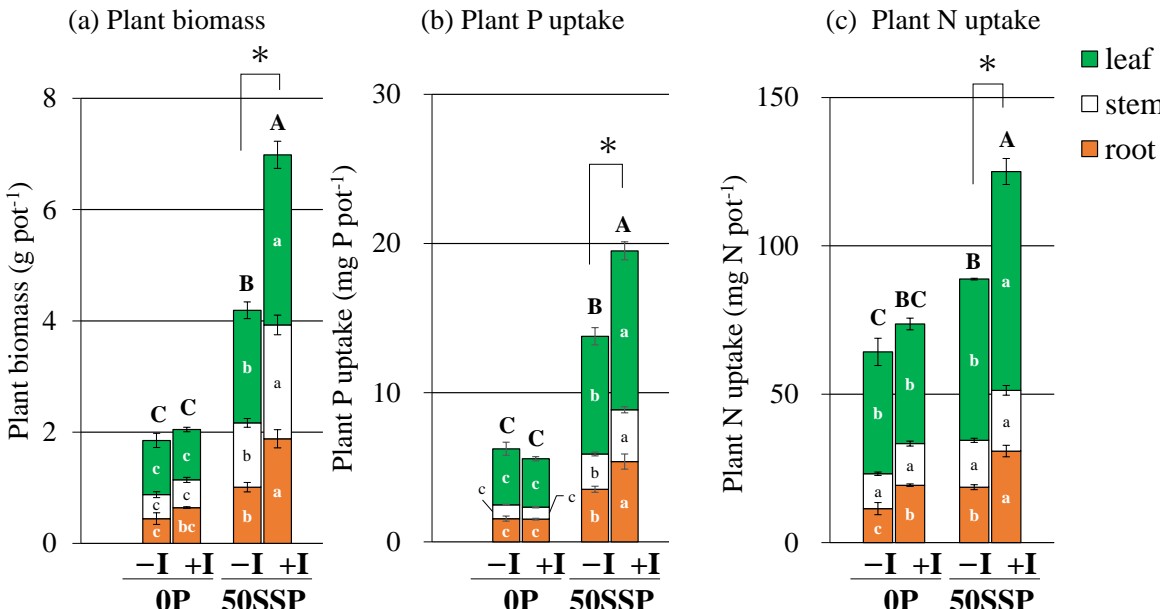

**Figure 1.** (**a**) Plant biomass, (**b**) Plant P uptake, (**c**) Plant N uptake after pot experiments. −I/+I and 0P/50SSP indicate *Rhizobium* inoculation and P fertilization, respectively. Bars indicate the standard error (*n* = 5). Lowercase letters indicate significant differences in total biomass, plant P uptake, and plant N uptake per pot between the plant parts under each treatment estimated using Tukey's posthoc test (*p* < 0.05). Uppercase letters indicate significant differences in total biomass, plant P uptake, and plant N uptake per pot in each treatment, according to Tukey's posthoc test (*p* < 0.05). * represents significant differences between −I and +I treatments under 0P and 50SSP treatments estimated using Student's *t*-test (*p* < 0.05).

**Table 2.** Summary of two-way analysis of variance for the effect of Inoculation ($I_{noc}$) and P fertilizer ($P_{fer}$) on plant biomass, P uptake, and N uptake.

| | Plant Biomass | | | | P Uptake | | | | N Uptake | | | |
|---|---|---|---|---|---|---|---|---|---|---|---|---|
| | Leaf | Stem | Root | Total | Leaf | Stem | Root | Total | Leaf | Stem | Root | Total |
| $I_{noc}$ | ** | * | *** | *** | * | NS | ** | * | ** | ** | *** | ** |
| | 8.2 | 17.1 | 22.8 | 17.0 | 5.6 | 16.0 | 8.0 | 9.3 | 7.3 | 9.2 | 45.4 | 17.7 |
| $P_{fer}$ | *** | *** | *** | *** | *** | NS | *** | *** | *** | *** | *** | *** |
| | 90.7 | 96.3 | 64.6 | 127.6 | 146.0 | 0.1 | 81.8 | 167.1 | 46.0 | 21.1 | 39.3 | 48.8 |
| $I_{noc} \times P_{fer}$ | ** | ** | * | ** | ** | NS | ** | ** | ** | NS | NS | * |
| | 10.7 | 12.6 | 9.0 | 12.9 | 11.7 | 1.0 | 8.6 | 14.8 | 8.5 | 1.1 | 2.1 | 6.1 |

$I_{noc}$ and $P_{fer}$ indicate the effect of *Rhizobium* inoculation and P fertilizer. NS, *, **, and *** indicate not significant, *p* < 0.05, *p* < 0.01, and *p* < 0.001, respectively, by two-way ANOVA. The value in the table indicates each F value.

### 3.2. Fractionated P of Rhizosphere and Bulk Soil

The amount of fractionated P (Resin-P, $NaHCO_3$-P, and NaOH-P) in the rhizosphere and bulk soil in each treatment is shown in Figure 2, and all the fractionated P values are shown in Table S2. P fertilization significantly increased $NaHCO_3$-Pi and NaOH-Pi, which accounted for approximately 40% and 30% of the added P, respectively. In 0P − I and 0P + I, no significant difference was observed between the rhizosphere and bulk soils for all the fractionated P contents. In contrast, in the 50SSP + I treatment, labile and moderately labile Pi, such as $NaHCO_3$-Pi and NaOH-Pi, were significantly higher in rhizosphere soil by 55% and 44%, respectively, compared to those in the bulk soil. However, these differences were not observed in 50SSP − I. In the case of labile and moderately labile Po, such as $NaHCO_3$-Po and NaOH-Po, both Po fractions were significantly higher in the rhizosphere soil than in the bulk soil, except for NaOH-Po in 50SSP + I.

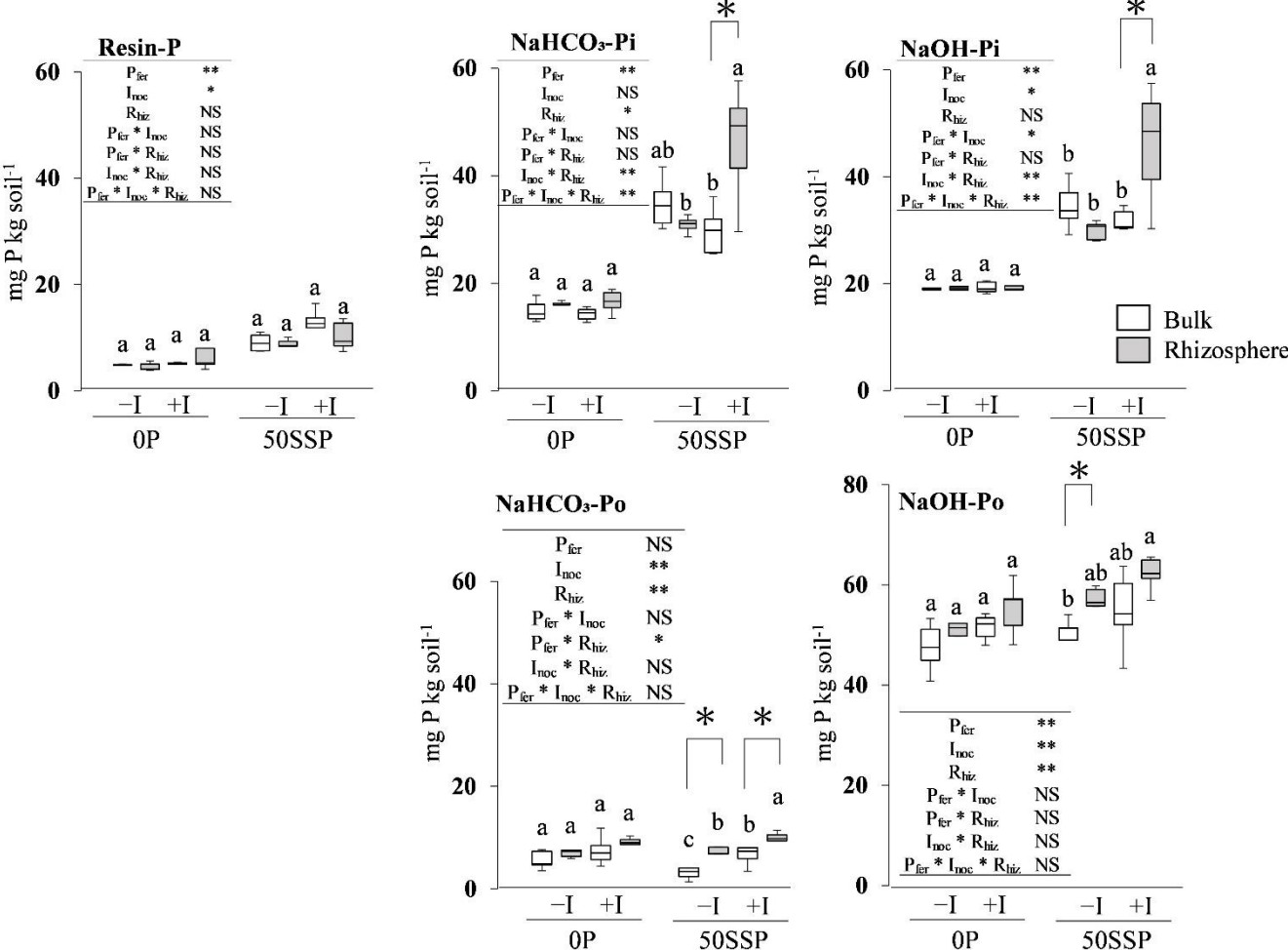

**Figure 2.** The amount of P fractionated by the modified Hedley's methods. −I/+I and 0P/50SSP indicate *Rhizobium* inoculation and P fertilization. Bars indicate the standard error (*n* = 5). Lowercase letters indicate the significant difference between the treatments, according to Tukey's posthoc test (*p* < 0.05). * represents significant differences between the −I and +I treatments under 0P and 50SSP treatments according to Student's *t*-test (*p* < 0.05). NS, *, and ** in inserted tables indicate not significant, *p* < 0.05 and *p* < 0.01, respectively, estimated using three-way ANOVA. $P_{fer}$, $I_{noc}$, and $R_{hiz}$ indicate the effect of P fertilizer, *Rhizobium* inoculation, and the rhizosphere, respectively.

The three-way ANOVA revealed that P fertilization and the *Rhizobium* inoculation treatments altered all the P fractions, except for $NaHCO_3$-Po by P fertilization and $NaHCO_3$-Pi. In addition, the interaction between P fertilization and *Rhizobium* inoculation and the interaction between P fertilization, *Rhizobium* inoculation, and the rhizosphere were observed only in $NaHCO_3$-Pi and NaOH-Pi.

### 3.3. Plant Root P Acquisition Characteristics

The amount of organic acid per root and pot in each treatment is shown in Figure 3 and in more detail in Tables S3 and S4. The sum of all organic acids per root did not differ significantly between +I and −I in both the P treatments nor between the 0P and 50SSP in either of the *Rhizobium* inoculation treatments (Figure 3a), indicating that *Rhizobium* inoculation or P fertilization did not affect the amount of organic acid per root. In the case of each organic acid (i.e., citric, malic, succinic, malonic, and oxalic acids) per root, no significant difference was observed in the *Rhizobium* inoculation. However, the amount of organic acid per pot, calculated by multiplying the organic acid per root by the root biomass, was higher under 50SSP + I than under 50SSP − I (ca. 130%, Figure 3b). In

contrast, no difference was observed for organic acid per pot in 0P + I and 0P − I. These data indicate that *Rhizobium* inoculation increased the total amount of organic acid per pot only in 50SSP.

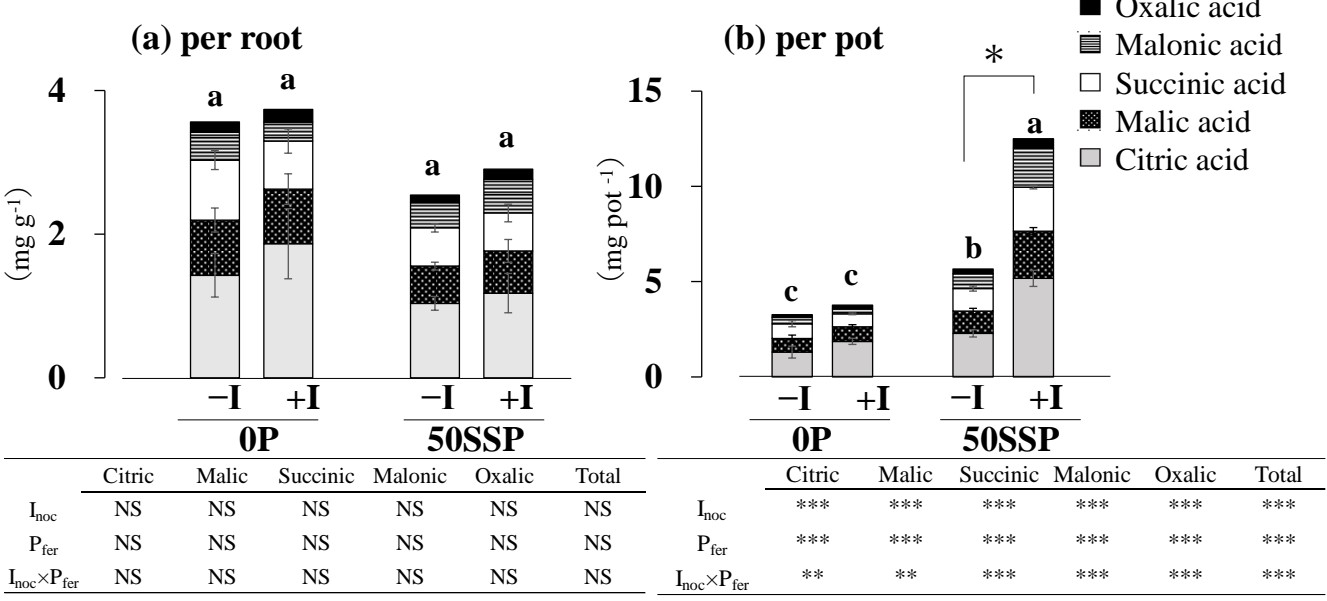

**Figure 3.** The amount of organic acid (**a**) per root, (**b**) per pot. −I/+I and 0P/50SSP indicate *Rhizobium* inoculation and P fertilization. Bars indicate the error bar (*n* = 5). Lowercase letters indicate the significant difference between the treatments, according to Tukey's posthoc test ($p < 0.05$). * represents significant differences between the −I and +I treatments under 0P and 50SSP treatments according to Student's *t*-test ($p < 0.05$). NS, *, **, and *** in inserted tables indicate not significant, $p < 0.05$, $p < 0.01$, and $p < 0.001$, respectively, estimated using two-way ANOVA; $I_{noc}$ and $P_{fer}$ indicate the effect of *Rhizobium* inoculation and P fertilizer, respectively.

The acid phosphatase activity per root and pot in each treatment is shown in Figure 4. The acid phosphatase activity per root under 0P + I was higher (~150%) than that under 50SSP + I. On the contrary, no difference was observed between 0P − I and 50SSP − I, indicating that *Rhizobium* inoculation combined with P fertilizer decreased the acid phosphatase activity per root. However, the acid phosphatase activity per pot was higher under 50SSP + I than under 0P−I. Furthermore, the two-way ANOVA analysis revealed that the P fertilization treatment altered both the acid phosphatase activity per root and pot, but not the *Rhizobium* inoculation.

### 3.4. Rhizosphere and Bulk Soil pH

The bulk and rhizosphere soil pH values for each treatment are presented in Table S5. The bulk soil pH in each pot was similar (5.20–5.51). The rhizosphere soil pH was higher than that of the bulk soil in all treatments, except for 50SSP + I, in which the rhizosphere soil pH (4.98) was lower than the bulk soil pH (5.22).

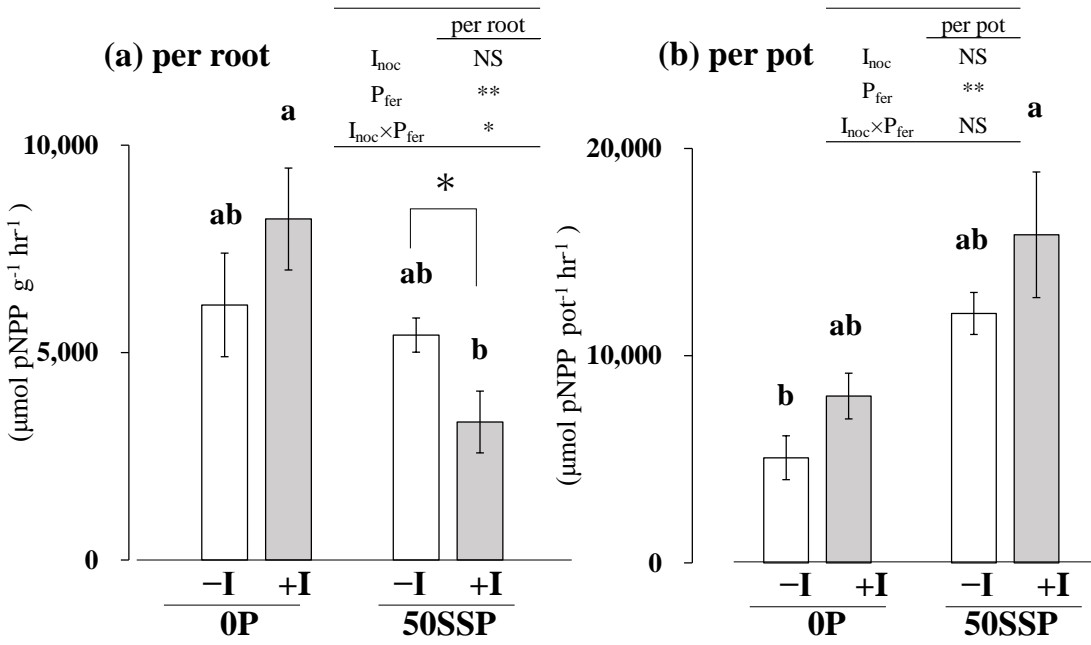

**Figure 4.** Acid phosphatase activity (**a**) per root and (**b**) per pot. −I/+I and 0P/50SSP indicate *Rhizobium* inoculation and P fertilization. Bars indicate the standard error (*n* = 5). Lowercase letters indicate the significant difference between the treatments, according to Tukey's posthoc test (*p* < 0.05). * represents significant differences between the −I and +I treatments under 0P and 50SSP treatments according to Student's *t*-test (*p* < 0.05). NS, *, and ** in inserted tables indicate not significant, *p* < 0.05, and *p* < 0.01, respectively, estimated using two-way ANOVA; $I_{noc}$ and $P_{fer}$ indicate the effect of *Rhizobium* inoculation and P fertilizer, respectively.

## 4. Discussion

We found a higher PUE, amount of organic acid per pot but not per root, and amounts of $NaHCO_3$-Pi and NaOH-Pi in the rhizosphere under 50SSP + I than under 50SSP − I, but not in the 0P treatments. These results supported our hypothesis that *Rhizobium* inoculation with P fertilization improves plant roots' P acquisition characteristics and rhizosphere soils' P conditions, resulting in a better PUE and crop growth in Ultisols.

### 4.1. Effect of the Rhizobium Inoculation with P Fertilization on the Rhizosphere Soil P Dynamics

We observed a larger amount of $NaHCO_3$-Pi and NaOH-Pi in the rhizosphere than in the bulk soil only in the case of the *Rhizobium* inoculation with P fertilization, that is, 50SSP + I. We considered that this accumulation of P was mainly derived from the fertilized P, but not from the native P of the soil, for the following reasons: (1) no similar P accumulation was observed in the rhizosphere in 0P, indicating that an excessive P dissolution of native soil P did not occur in 0P + I and 0P − I; and (2) both the total amounts of organic acid per pot and the total plant P uptake were increased by the *Rhizobium* inoculation only in 50SSP + I, indicating that the solubilization and accumulation of fertilizer P in the rhizosphere occurred only in 50SSP + I. Previous studies have shown that such accumulation of P in the rhizosphere could result from the interactions between P ions and substantial organic acids from the root [51]. Li et al. [52] reported that P-mobilizing legumes increased labile P in the rhizosphere by secreting organic acids and phosphatases. Sugihara et al. [20] also observed fertilizer P accumulation in pigeon pea rhizospheres grown in Tanzanian cropland soil. Reportedly, a high soil P availability enhances plant growth, leading to increased carbon assimilation into the roots and exudate, resulting in larger root systems or exudates and thus better plant P acquisition (positive feedback) [53,54]. In this study, relatively smaller $NaHCO_3$-Pi and NaOH-Pi concentrations were detected in the rhizosphere soil than in the bulk soil in 50SSP − I. This indicates that the P fertilization treatment alone

cannot accumulate P in the rhizosphere; however, *Rhizobium* inoculation with appropriate P fertilization improves PUE through the accumulation of fertilizer P in the rhizosphere.

### 4.2. Effect of the Rhizobium Inoculation with P Fertilization on Plant P Acquisition

The present study showed that the organic acid concentration per pot in 50SSP +I was ~230% higher than that in 50SSP − I, indicating the apparent synergistic effect of Rhizobium inoculation and P fertilization on plant P acquisition via the exudate of organic acid (Figure 3b). There was no apparent difference in the organic acid content per root for the Rhizobium inoculation treatments in 50SSP (Figure 3b). In comparison, there was a clear increase in root biomass (160%) with the Rhizobium inoculation, with an increase in the weight of the root nodule (158%) only in the 50SSP treatment (Figure 1a). Generally, P fertilizer stimulates plants to increase their root biomass [55], while it also increases $N_2$ fixation by Rhizobium, which requires P as the primary source of $N_2$ fixation energy [56,57]. Rhizobium inoculation can also increase rhizobial colonies in the rhizosphere and promote $N_2$ fixation [58], causing efficient root mass increase. These results suggest that the increased root biomass by Rhizobium inoculation with P fertilization mainly contributes to the efficient plant P acquisition capacity in 50SSP + I rather than improving the root P acquisition capacity. These results were also consistent with the results of the rhizosphere P dynamics, which showed the accumulation of labile and moderately labile Pi in the rhizosphere only in 50SSP + I. In the current study, all measured organic acids, including citric, malic, succinic, malonic, and oxalic acid, were evenly increased per pot by Rhizobium inoculation with P fertilization. Citric acid was the most abundant, accounting for 42% of the total amount (Figure 3b, Table S4). Many studies have reported the significance of citric acid in solubilizing soil P or fertilized P [59–61], while Oburger et al. [62] also found that a mixture of various organic acids, such as citric, malic, oxalic, and malonic acids, is important for efficient P desorption from the soil. In addition, Otani et al. [16] also observed that malonic acid was the major component that solubilized the Fe-associated P in the soil via the pigeon peas. In our study, the malonic acid in 50SSP + I accounted for 17% of the total amount. Thus, the increased citric acid and malonic acid may contribute to efficient P solubilization and better PUE in 50SSP + I, although further study is necessary. In addition, the impact of sole Rhizobium inoculation on organic acid in 0P is not clear, indicating the necessity of P for the N-fixing activity of plant-Rhizobium symbiosis [31,63,64].

We also observed that P fertilization increased the acid phosphatase activity per pot and decreased it per root (Figure 4). These results indicate that the increased root biomass in 50SSP contributes to plants' P acquisition capacity via the exudate of acid phosphatase, similar to organic acid. This is consistent with the increase in labile Po in the rhizosphere in both 50SSP − I and 50SSP + I but not in 0P − I and 0P + I (Figure 2). Rejmánková and Macek [65] also found that P fertilization decreased the acid phosphatase activity of the roots in three different plants at graded P concentrations. Krishnappa and Aftab Hussain [66] also observed similar results, such as decreased acid phosphatase activity in pigeon pea roots under high P conditions. In our study, Rhizobium inoculation did not affect acid phosphatase activity either per root or per pot, indicating that Rhizobium inoculation had little impact on the acid phosphatase activity of roots in either 0P or 50SSP. Thus, we considered that the increased organic acid exudate mainly contributed to the solubilization and accumulation of rhizosphere P and better PUE in 50SSP + I, rather than the increased acid phosphatase activity [51]. Ae et al. [67] also reported that pigeon peas solubilized moderately labile P, such as Fe-associated P, mainly through organic acids.

## 5. Conclusions

In this study, we observed that Rhizobium inoculation increased the PUE of pigeon peas (25.5% in 50SSP + I and 14.4% in 50SSP − I). The revealed mechanism for higher PUE was as follows: (1) Rhizobium inoculation with P fertilization first stimulates the P acquisition capacity of pigeon peas by increasing the root biomass and the secretion

of organic acid; (2) the increased plant P acquisition contributes to the solubilization or accumulation of fertilizer P in the rhizosphere; and (3) the improved rhizosphere P condition improves the PUE of the pigeon peas. Our results suggest that the root biomass, such as the amount of organic acid per pot, is likely to be more important in improving plants' P acquisition capacity. Therefore, Rhizobium inoculation with P fertilization could be an effective technique to improve the PUE of pigeon peas in strongly weathered tropical soil, where the PUE is generally low [68,69].

**Supplementary Materials:** The following supporting information can be downloaded at: https://www.mdpi.com/article/10.3390/agronomy12123149/s1, Figure S1: The condition at harvest (screening tests for selection of *Bradyrhizobium* inoculant by 30 days pot experiment); Figure S2: Plant condition at harvest and the total weight of *rhizobium* root nodules; Table S1: Plant P and N concentration after pot experiment; Table S2: Amount of fractionated and total soil P after the pot experiment; Table S3: The amount of organic acid per root; Table S4: The amount of organic acid per pot; Table S5: pH after pot experiment.

**Author Contributions:** Conceptualization: S.S.; data curation: S.Y., S.O., N.O.-O. and S.S.; investigation: S.Y., N.D.M., H.T. and S.S.; methodology, S.O., N.O.-O. and S.S.; project administration: S.S.; writing—original draft: S.Y.; writing—review and editing, S.O., N.D.M., N.O.-O., H.T. and S.S. All authors have read and agreed to the published version of the manuscript.

**Funding:** This research was funded by Japanese Society for the Promotion of Science KAKENHI; grant Numbers #17H06171, #18KK0185, and #22H04882 financially supported this work.

**Acknowledgments:** We thank Suzuki and other staff members at Tokyo University of Agriculture and Technology for supporting in the *Rhizobium* inoculation techniques.

**Conflicts of Interest:** The authors declare no conflict of interest.

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
