# Peer review of "Rhizobium Inoculation Improved the Rhizosphere P Dynamics and P Uptake Capacity of Pigeon Pea Plants Grown in Strongly Weathered Soil Only under P Fertilized Conditions"

_agronomy, doi:10.3390/agronomy12123149_

Round 1
Reviewer 1 Report
Dear Authors,
congratulations on a well-written, substantively valuable manuscript. I only have a few minor comments of an editorial nature:
L 105: please provide the name of the pigeon pea cultivar.
L 110-111: “One day later, four seeds per pot were planted, and rhizobia solution was applied to each seed again (3.0 × 109 cell per seed).” - please explain how "rhizobia solution was applied to each seed again".
Latin names should be written in italics, e.g. Cicer arietinum in L 490, Albizia falcataria in L 498, Cajanus cajan in L 569.
I wish the authors further fruitful work.
Yours sincerely,
Reviewer
Reviewer 2 Report
The work by Yamamoto and colleagues entitled ¨Rhizobium inoculation improved the rhizosphere P dynamics and P uptake capacity of pigeon pea plants grown in strongly weathered soil only under P fertilized conditions¨, conducts research on the ability of Rhizobium to stimulate P uptake in pigeon pea plants.
In general, the work is of very good quality, and the introduction is complete and justifies the work very well. The only criticism I would have would be the lack of novelty, as there are countless similar works in the literature, but this work is worth publishing for its quality of writing and how well it is done.
In M&M the tests carried out are described in detail.
The results are enough to complete the work history and the graphics are of good quality and very explicit.
The discussion is also good.
Only in the Conclusion part is there a somewhat speculative statement, regarding the mechanism that should be suggested, and it could say: 1) Rhizobium inoculation with P fertilization would first stimulates the P acquisition capacity....
in 2) the increased plant P acquisition would contribute...
